# Grouping of Ankyloglossia According to Coryllos Anatomical Classification and Follow-Up Results for Breastfeeding: Single-Center, Cross-Sectional Study

**DOI:** 10.3390/children9121860

**Published:** 2022-11-30

**Authors:** Mehmet Ali Narsat, Abdulvehhap Beygirci, Gökçen Tuğçe Özdönmez, Eren Yıldız

**Affiliations:** 1Department of Pediatric Surgery, Faculty of Medicine, Kastamonu University, Kastamonu 37150, Turkey; 2Department of Pediatrics, Kastamonu Training and Research Hospital, Kastamonu 37150, Turkey; 3Department of Pediatrics, Faculty of Medicine, Kastamonu University, Kastamonu 37150, Turkey

**Keywords:** ankyloglossia, breastfeeding, Coryllos, frenotomy, infant, lingual frenulum, newborn, tongue tie

## Abstract

Ankyloglossia is a condition of limited tongue mobility caused by a short lingual frenulum. The diagnosis and treatment of ankyloglossia are still controversial. The main clinical problems encountered during breastfeeding are difficulty in sucking and its clinical reflections. This study aims to evaluate the infant population born with ankyloglossia and to determine the results of frenotomy. We conducted an observational, cross-sectional study among infants born in a tertiary hospital. We included all infants born between 1 January and 30 June 2022. The neonatal follow-up protocol for ankyloglossia was determined before the defined dates, and data were recorded during the screening period. The recorded data were retrospectively collected from the files. Within six months, 705 infants were born. Due to additional problems and other conditions that prevent breastfeeding, evaluable data of 207 (29.3%) infants could not be provided. Of the remaining 498 infants, 234 (33.2%) had ankyloglossia. While none of the infants without ankyloglossia had a breastfeeding problem after appropriate training, 28.6% of the ankyloglossia group had a breastfeeding problem (*p* < 0.001). The need for frenotomy differed significantly between Coryllos groups (*p* < 0.001). Breastfeeding was unsuccessful before frenotomy in 12 Coryllos type-1 patients, and all had difficulty in sucking. Frenotomy was performed within the three-month follow-up period in all patients with complaints of inability to firmly grasp the breast, nipple slipping from the mouth, and nipple biting during the first 24 h. In terms of breastfeeding problems, regardless of the anatomical typology, frenotomy can be performed safely in early life with successful results. If deficiencies or difficulties in breastfeeding are noticed in ankyloglossia patients even at the first control, frenotomy should be recommended in clinical conditions.

## 1. Introduction

Ankyloglossia or tongue tie is defined as limited tongue mobility caused by a short lingual frenulum [1]. The lingual frenulum is a submucosal or membranous connective tissue component at the base of the tongue [2]. Abnormal development or incomplete dissociation of this ligament in congenital developmental stages results in ankyloglossia [3].

The diagnosis and treatment of ankyloglossia are still controversial [4,5,6,7]. According to the methods used for diagnosis, the incidence of ankyloglossia varies between less than 1% and 46.3% [8,9,10]. Sometimes, the diagnosis may go unnoticed, and patients encounter treatment delays and clinical problems; in other instances, the patients may receive unnecessary interventions [11]. The main clinical problems encountered during breastfeeding are difficulty in sucking and its clinical reflections. Intervention is recommended as a consensus in patients with significant tongue movement limitation [1]. There are often no breastfeeding problems even in this group of patients [1]. Even if breastfeeding is successful, problems may occur in tooth development, speech, and normal anatomical and functional development of the mouth and jaw structure at later ages [1,12].

Many different methods have been described for the diagnosis of ankyloglossia [1,4,6]. Among these methods, there are methods that evaluate anatomical and functional properties. Coryllos ankyloglossia classification is a method that clinicians can easily evaluate based on anatomical appearance [13]. Coryllos typing is frequently used in studies to describe the common anatomical appearance.

There are different methods that can be applied for treatment [1]. There are studies in the literature in which surgical procedures such as frenoplasty were applied [1,14]. However, there are also centers where frenotomy is performed [1,14]. Recurrence is less likely in major procedures performed under anesthesia. Frenotomy, which is an easily applicable method in the outpatient clinic, has a high probability of recurrence, but it does not require anesthesia, it is more practical, and the major complications related to frenotomy are much less; therefore, it is preferred. In the literature, there is not a single correct treatment method recommended as a common opinion yet [1].

This study aimed to characterize the Kastamonu Training and Research Hospital infant population according to the presence of ankyloglossia, characterize the types of ankyloglossia, decide whether or not it is clinically significant, and present the results of frenotomy procedures.

## 2. Materials and Methods

### 2.1. Study Plan

We conducted an observational and cross-sectional study of infants born in a tertiary hospital that has approximately 1500 deliveries per year and serves a population of roughly 300,000. We included all infants born at Kastamonu Training and Research Hospital between 1 January and 30 June 2022. The study algorithm was determined before the defined dates, and data were recorded during the screening period. The recorded data were retrospectively collected from patient files (Figure 1).

### 2.2. Neonatal Follow-Up Protocol for Ankyloglossia

Each infant was subjected to neonatal examination by a pediatrician and pediatric surgeon. Coryllos groupings were noted in the neonatal examination records [13] (Table 1). Mothers of all infants born in Kastamonu Training and Research Hospital were given proper breastfeeding education and educational materials within national and international standards. Nutritional characteristics were recorded on the 1st, 7th, 28th, 60th, and 90th days postpartum by conducting face-to-face interviews with the mother, monitoring at least one feeding, and re-examination. Pain during breastfeeding, bruises and skin sores on the breast, and mastitis were evaluated and recorded. The infants’ poor latch, inability to grasp the breast deeply, nipple slipping out of the mouth, nipple biting, suction power, suction amount, weight gain, and frequent and short-term partial breastfeeding were also recorded [13].

Other problems (problems related to prematurity, swallowing dysfunctions, choanal atresia, swallowing disorders due to neurological developmental disorders, anatomical problems except ankyloglossia, etc.) that may cause feeding problems in infants were re-evaluated. Mothers received education in proper nutrition with breast milk one more time. Frenotomy was performed on babies with ankyloglossia who did not have any additional problems that would cause nutritional problems in clinical conditions. Infants who underwent frenotomy were followed up until they were 90 days old.

### 2.3. Frenotomy Procedure

All frenotomies were performed by pediatric surgeons in clinical conditions. There was no local or general anesthetic used. The infants were immobilized. The frenotomy was carried out by raising the tongue and cutting the frenulum up to the base of the tongue with blunt-tipped Metzenbaum scissors, allowing for full exploration with a threaded sublingual ligament retractor. After the frenotomy, the patients were kept under observation for two hours. A health professional observed infants during their first feeding.

### 2.4. Statistical Analysis

Categorical variables (sex, method of delivery, prematurity, presence of ankyloglossia, clinically significant ankyloglossia, and Coryllos classification) were expressed as frequencies and percentages. We compared the populations with and without ankyloglossia, and with and without frenotomy. Qualitative variables were compared with Pearson’s chi-square test and Fisher’s exact test. Student’s T test was used for quantitative evaluations. Significance was set as *p* < 0.05. To perform statistical analyses, we used version 22 of the IBM SPSS program.

## 3. Results

### 3.1. The Study Group in General

Within six months, 705 infants were born. Due to additional problems (problems related to prematurity, swallowing dysfunctions, choanal atresia, swallowing disorders due to neurological developmental disorders, anatomical problems except ankyloglossia, etc.) and other conditions that prevent breastfeeding, evaluable data of 207 (29.4%) infants could not be provided. Of the remaining 498 infants, 234 (33.2%) had ankyloglossia. Of the patients, 124 (52.0%) were boys, and 110 were girls (47.0%). Compared to 46 (19.6%) infants born by cesarean section, 188 (80.3%) were born by spontaneous vaginal delivery. When the study group (*n* = 498) was divided into groups with ankyloglossia (*n* = 234) and without ankyloglossia (*n* = 264), they were statistically similar in terms of demographic characteristics, delivery type, birth weight, and gender (*p* > 0.05). While none of the infants without ankyloglossia had a breastfeeding problem after appropriate breastfeeding education, 67 (28.6%) of the ankyloglossia group had a breastfeeding problem (*p* < 0.001).

### 3.2. Ankyloglossia Baby Group

Coryllos type 3 was the most common (70.5%) tongue-tie appearance. Frenotomy was performed in 67 patients due to clinical breastfeeding difficulties caused by ankyloglossia. The need for frenotomy differed significantly between Coryllos groups (*p* < 0.001) (Table 2).

The nutritional characteristics of the patients are given in Table 3. Breastfeeding was unsuccessful before frenotomy in 12 Coryllos type-1 patients, and all had difficulty in sucking; moreover, all Coryllos type-1 patients underwent frenotomy during the follow-up period. In addition, frenotomy was performed in all patients who complained of inability to grasp the breast deeply in the first 24 h, nipple slipping out of the mouth, and nipple biting (*n* = 67) during the 3-month follow-up period. On the other hand, babies who could latch from the start (*n* = 167) did not need frenotomy due to breastfeeding problems in the following period, regardless of the Coryllos type.

The mothers of infants with ankyloglossia did not experience pain while breastfeeding, bruises and skin sores on the breasts, or mastitis.

There were no early or late surgical complications and no need for re-frenotomy in patients who underwent frenotomy. During the 90-day follow-up, the patients who underwent frenotomy did not experience feeding problems.

## 4. Discussion

The incidence of ankyloglossia in our study group was 33.2%. This rate is considerably higher than the 0.3% ankyloglossia rate in the study conducted by Çetinkaya et al. in 2011 examining general oral cavity anomalies [8]. In addition, different rates are seen in various studies, ranging from less than 1% to 46.3% [8,9,15,16]. These differences are due to varying diagnostic methods and criteria. For example, in the study conducted by Maya-Enero et al., a similarly planned study to ours, the rate was 46.3% [9]. Maya-Enero et al. reported symptoms in 70.2% of patients [9]. In our study, the rate of symptomatic patients was 28.7%. The discrepancy in this result alone reveals that there is still no definite opinion. Especially in Coryllos type-3 and type-4 patient groups, the rates are higher than in other studies [8,15,16]. We determined the necessity of surgical intervention according to clinical features rather than the scales used [17,18,19,20]. Our results are consistent with the questions Manteli evaluated regarding function [19]. It is not certain that breastfeeding problems will occur when ankyloglossia is present. However, frenotomy is beneficial if ankyloglossia is present in patients with breastfeeding problems.

In our study, we followed the patients for 90 days. This period is not enough time to reveal all the consequences of ankyloglossia. However, the main finding in our study is that the complaints of babies who cannot latch, experience nipples slipping out of their mouths, and nipple biting caused by ankyloglossia within the first 24 h continue regardless of type, and that they need frenotomy for comfortable breastfeeding in follow-ups. Although our frenotomy results seem better compared to the literature, there is still not enough information about problems other than breastfeeding that may arise for a longer time [21,22,23]. It has been shown that the breastfeeding problems of the patients we applied frenotomy with control appointments were eliminated.

Although some researchers asked mothers about maternal difficulties with breastfeeding, we did not observe such problems [1,24]. We can solve the problems of patients who bite nipples and cannot latch on effectively in the neonatal period using our bedside frenotomy procedure before breast damage ensues. Similarly, Messner’s article of consensus in 2020 and Emond’s study in 2014 demonstrate that early frenotomy procedures prevent maternal problems [1,25].

In our ankyloglossia patients, breastfeeding problems disappeared after frenotomy. We had no complications in the frenotomy procedure, which has a low complication rate in the literature [21,22,23]. Frenotomy should not be avoided in infants with symptoms since it contributes to breastfeeding, and the complication rates are acceptable.

In clinical conditions, frenotomy can be performed more easily than other methods described [26]. As Coryllos mentioned in his publication in 2004, he used local anesthesia during the frenotomy technique [13]. We did not require the use of local anesthetics in our procedure. Similarly, in the study conducted by Sethi et al., they did not use local or systemic anesthetics for infants [27]. Topical anesthetics may increase patient comfort in children [28]. If ankyloglossia is not noticed in patients during infancy until preschool age or later, the procedure might need to be performed under general anesthesia [28].

The main limitation of our study is the short follow-up period since the patients may have some clinical complaints of ankyloglossia, especially after starting to speak. Despite the extensive literature, there is no definitive diagnosis or treatment method for ankyloglossia. Therefore, like all other studies on ankyloglossia, our study has similarities with others and a different approach to the subject. Differences in diagnostic criteria and treatment methods limit the reproducibility of the study.

## 5. Conclusions

With screening programs, it will be seen that ankyloglossia is more common than the general literature shows.Frenotomy can be performed without delay in patients with ankyloglossia and breastfeeding problems.Frenotomy give successful results.If difficulties in breastfeeding are noticed in ankyloglossia patients even at the first control, the frenotomy procedure should be recommended in clinical conditions.

## Figures and Tables

**Figure 1 children-09-01860-f001:**
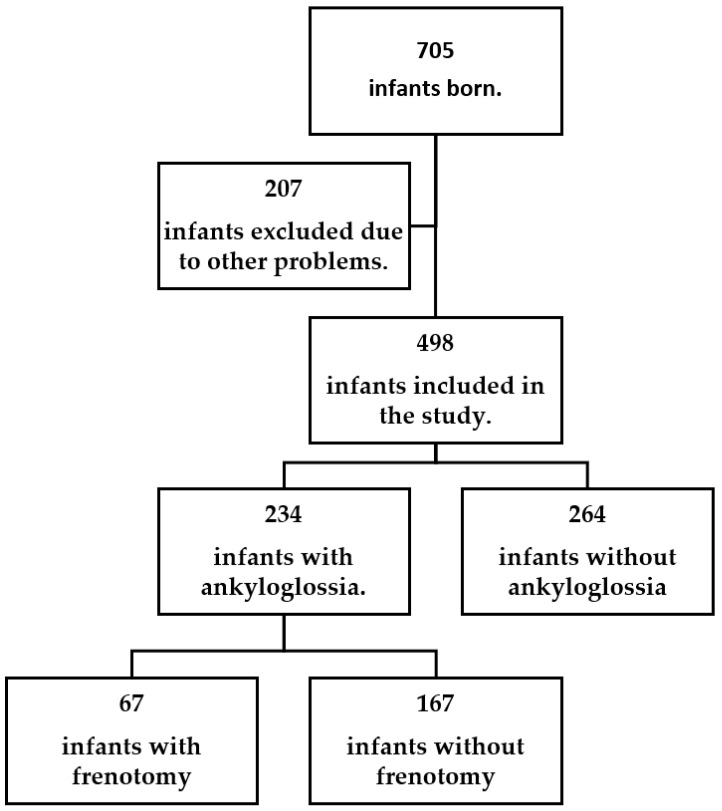
Patient diagram.

**Table 1 children-09-01860-t001:** Types of Coryllos ankyloglossia [13].

Type 1: Insertion of the frenulum to the tip of the tongue
Type 2: Insertion of the frenulum slightly (2 to 4 mm) behind the tip of the tongue
Type 3: Thickened frenulum attached to the mid-tongue and the middle of the floor of the mouth, usually tighter and less elastic
Type 4: Thick, shiny, and very inelastic submucosal frenulum that restricts movement at the base of the tongue

**Table 2 children-09-01860-t002:** Coryllos groups and frenotomy distribution.

	Frenotomy		Total	*p*
	**Done**	**Not Done**		
Coryllos type 1	14 (6.1%)	0	14 (6.1%)	<0.001
Coryllos type 2	32 (13.7%)	12 (5.1%)	44 (18.8%)	
Coryllos type 3	15 (6.4%)	150 (64.1%)	165 (70.5%)	
Coryllos type 4	6 (2.5%)	5 (2.1%)	11 (4.6%)	
Total	67 (28.7%)	167 (71.3%)	234 (100%)	
*Chi Square*				

**Table 3 children-09-01860-t003:** Nutritional characteristics (*n* = 234).

		Type 1	Type 2	Type 3	Type 4	Total
Frenotomy Procedure	Day 1	12	8	3	0	23
	Day 7	2	24	12	4	42
	Day 28	0	0	0	2	2
	Day 60	0	0	0	0	0
	Day 90	0	0	0	0	0
Unable to Breastfeed	Day 1	12	8	3	0	23
	Day 7	0	0	0	0	0
	Day 28	0	0	0	0	0
	Day 60	0	0	0	0	0
	Day 90	0	0	0	0	0
Weak Grasp	Day 1	12	8	3	0	23
	Day 7	0	0	0	0	0
	Day 28	0	0	0	0	0
	Day 60	0	0	0	0	0
	Day 90	0	0	0	0	0
Poor Latch	Day 1	14	32	15	6	67
	Day 7	2	24	12	6	34
	Day 28	0	0	0	2	2
	Day 60	0	0	0	0	0
	Day 90	0	0	0	0	0
Nipple Slipping Out of the Mouth	Day 1	14	32	15	6	67
	Day 7	2	24	12	6	34
	Day 28	0	0	0	2	2
	Day 60	0	0	0	0	0
	Day 90	0	0	0	0	0
Nipple Biting	Day 1	14	32	15	6	67
	Day 7	2	24	12	6	34
	Day 28	0	0	0	2	2
	Day 60	0	0	0	0	0
	Day 90	0	0	0	0	0
Weak Suction	Day 1	14	32	15	6	67
	Day 7	2	24	12	6	34
	Day 28	0	0	0	2	2
	Day 60	0	0	0	0	0
	Day 90	0	0	0	0	0
Weight Loss	Day 1	0	0	0	0	0
	Day 7	0	0	0	0	0
	Day 28	0	0	0	2	2
	Day 60	0	0	0	0	0
	Day 90	0	0	0	0	0
Frequent-Short Feeding Durations	Day 1	14	32	15	6	67
	Day 7	2	24	12	6	34
	Day 28	0	0	0	2	2
	Day 60	0	0	0	0	0
	Day 90	0	0	0	0	0

## Data Availability

Data supporting the study are available from the corresponding author upon request.

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
