# Peer review of "Grouping of Ankyloglossia According to Coryllos Anatomical Classification and Follow-Up Results for Breastfeeding: Single-Center, Cross-Sectional Study"

_children, 2022, doi:10.3390/children9121860_

Round 1

Reviewer 1 Report

There are too much unnecessary controversies regarding treating ankyloglossia in infants despite obvious clinical findings and problems with breastfeeding. This paper is another proof that early treatment of the tongue tie is an important aspect of neonatal care. I hope more will be done in this field shortly. 

Author Response

Dear Reviewer 1,

Thank you very much for such a positive feedback.

Best regards

Reviewer 2 Report

The paper entitled “Grouping of Ankyloglossia According to Coryllos Anatomical Classification and Follow-Up Results for Breastfeeding: Single Center, Cross-Sectional Study” is a contribute that  aims  to evaluate the newborn population born regarding ankyloglossia and to determine the results of frenotomy. The work  provides  original data and it could be considered of interest for the readers. However before it could be considered suitable for publication requires several corrections.

INTRODUCTION

It is too sintetic and do not provides all the information necessary to understand the scientific background, the knowledge gap and the objectives of the study. This part of the paper need a complete revision.

MATERIAL AND METHODS

The scientific methodology used were described in a clear and exhaustive manner. There is no iconography related to the methodology used in the research. In this part of the paper should be added some figures. The statistical analysis used to evaluate the results unsufficient.

RESULTS AND

The results are described in a precise and detailed manner; graphical representation is well executed and allows a faster understanding of the results achieved in the study.

DISCUSSION

The discussion of the results is on the whole well articulated ; clinical relevance of the results should be emphasized more.

CONCLUSION

Conclusions are limited to a synthetic summary of the results obtained; this section must be revised and report preferably with a bulleted list, only the key results of the study.

Author Response

Dear Reviewer 2, 

Thank you for your nice comments and support. Adjustments were made in line with your suggestions and the article was evaluated.

INTRODUCTION

The introduction has been expanded and rewritten. The scientific basis of the topics highlighted in the study has been prepared.

MATERIAL AND METHODS

In the method section, patient diagram and numbers are added as figures. Statistical methods written without appendices have been added.

RESULTS

We are honored that you found our presentation style understandable.

DISCUSSION

The importance of frenotomy and its early application has been repeatedly added to the discussion.

CONCLUSION

In line with your suggestions, the conclusion section has been rewritten.

Sincerely.

Reviewer 3 Report

Dear authors,

I would like to thank you for the opportunity to review your manuscript. This research aims to evaluate the newborn population born regarding ankyloglossia and to determine the results of frenotomy, through a cross-sectional study.

## General comments to authors: This cross-sectional study presents an interesting research topic that could be translated to clinical practice. This study is a suitable option to add more knowledge about the ankyloglossia and therapeutic options to improve quality of life. However, before considering for publication manuscript should address some issues. I hope my comments could help authors to enrich their manuscript.

Abstract:

# Comment 1: Lines 24-28. Results from quantitative analysis have to appeared in the abstract.

Introduction:

# Comment 1: Please, to ease readers provide some information of Coryllos Anatomical Classification.

# Comment 2: Please, considering frenotomy as an important part of the authors’ study, it should be mentioned in the introduction section.

Methods:

# Comment 1: Line 56. What do the authors mean with the study algorithm? This expression also appeared in the abstract and in section 2.2.

# Comment 2: Line 74. Please, authors should specify what other problems were re-evaluated.

# Comment 3: Line 91. How quantitative data were compared?

# Comment 4: As a randomized clinical trial did authors followed the CONSORT guidelines?

Results:

# Comment 1: Please, include a CONSORT flow chart and explain why participants dropout from study. It is an important consideration specified in the CONSORT guideline.

# Comment 2: Please, include in the first part of results, the number of participants, main characteristic and write if there were difference between groups at baseline.

# Comment 3: To ease readers to understand results the authors should provide a narrative interpretation of results. Also, I could help if authors divide the results in sections 

# Comment 4: In the introduction section authors specified that one of the aims was to set the prevalence. However, only information about incidence throughout the text appeared. Since prevalence and incidence are not the same terms and concepts, authors should clarify this major issue.

# Comment 5: Line 90. After reading the manuscript the comparison between babies with ankyloglossia and without it, seems to be scares. Could authors provide wider information, please?

# Comment 6: Please authors, considered changing throughout the text the expression anlyloglosia patients by babies/newborns with ankyloglossia.

Discussion:

# Comment 1: Line 163-165. Regarding authors wrote this statement in the abstract “In terms of breastfeeding problems, regardless of the anatomical typology, frenotomy can be performed safely in the early life with successful results.” and subsequently the following in the discussion “In addition, the effect of frenotomy could not be fully demonstrated since it was performed on all patients with clinical problems, and the nursing issues of the patients on who we performed frenotomy were fixed with follow-up appointments”, it seems that there is a spin in abstract. Authors should solve this issue.

# Comment 2: Line 169. Please authors, clarify this important sentence.

Conclusion:

# Comment 1: Please, authors should consider to change conclusion section to be in line with their findings.

Author Response

Dear Reviewer 3,
Thank you for your careful review. We are sure that our article has become more perfect in line with the opinions and suggestions of you and other referees.
We have tried to answer your comments one by one below, and we are grateful for your valuable contributions.

Abstract:

# Comment 1: Lines 24-28. Results from quantitative analysis have to appeared in the abstract.

There are no statistical and scientific quantitative analysis results in the study. The result of the qualitative evaluation we made according to the Coryllos groups and ankyloglossia is also included in the summary section.

Introduction:

# Comment 1: Please, to ease readers provide some information of Coryllos Anatomical Classification.

Necessary additions have been made in line with your valuable comment.

# Comment 2: Please, considering frenotomy as an important part of the authors’ study, it should be mentioned in the introduction section.

Necessary additions have been made in line with your valuable comment.

Methods:

# Comment 1: Line 56. What do the authors mean with the study algorithm? This expression also appeared in the abstract and in section 2.2.

It is our follow-up method in our clinic, which we specify as the working algorithm. It has been replaced with the phrase "the neonatal follow-up protocol for ankyloglossia", which we think is more understandable.

# Comment 2: Line 74. Please, authors should specify what other problems were re-evaluated.

Problems related to prematurity, swallowing dysfunctions, choanal atresia, swallowing disorders due to neurological developmental disorder, anatomical problems except ankyloglossia, etc.
Added method section.

# Comment 3: Line 91. How quantitative data were compared?

Student T test was used for quantitative evaluations. Added to the method section.

# Comment 4: As a randomized clinical trial did authors followed the CONSORT guidelines?

The study was not randomized. All of our patients were evaluated and divided into groups as babies who experienced and did not experience the described distress. For this reason, the CONSORT guidelines were not followed.
We are planning a randomized study after this study, which can form a basis for randomized studies. Thanks again for your suggestion.

Results:

# Comment 1: Please, include a CONSORT flow chart and explain why participants dropout from study. It is an important consideration specified in the CONSORT guideline.

Flow chart has been added. The flowchart does not meet all the requirements of the CONSORT guideline, as there is no randomized or no frenotomy group. The patients who were excluded from the study group were either not suitable for evaluation due to additional problems or had significant obstructive sucking conditions other than ankyloglossia. The explanation has been added to the findings.

# Comment 2: Please, include in the first part of results, the number of participants, main characteristic and write if there were difference between groups at baseline.

The section has been rearranged by adding the necessary information.

# Comment 3: To ease readers to understand results the authors should provide a narrative interpretation of results. Also, I could help if authors divide the results in sections 

In line with the suggestions, the points that should be emphasized in the table were added as text in the findings section. The results section was divided into two parts as the results of the general study group and the infant group with ankyloglossia.

# Comment 4: In the introduction section authors specified that one of the aims was to set the prevalence. However, only information about incidence throughout the text appeared. Since prevalence and incidence are not the same terms and concepts, authors should clarify this major issue.

Incorrect use of "determine its prevalence" has been removed.

# Comment 5: Line 90. After reading the manuscript the comparison between babies with ankyloglossia and without it, seems to be scares. Could authors provide wider information, please?

More descriptive additions were made to the results section.

# Comment 6: Please authors, considered changing throughout the text the expression anlyloglosia patients by babies/newborns with ankyloglossia.

While talking about the first 28-day-old baby, the words newborn were used when talking about babies after 28 days. However, to avoid confusion, the words in the entire text have been changed to infant.

Discussion:

# Comment 1: Line 163-165. Regarding authors wrote this statement in the abstract “In terms of breastfeeding problems, regardless of the anatomical typology, frenotomy can be performed safely in the early life with successful results.” and subsequently the following in the discussion “In addition, the effect of frenotomy could not be fully demonstrated since it was performed on all patients with clinical problems, and the nursing issues of the patients on who we performed frenotomy were fixed with follow-up appointments”, it seems that there is a spin in abstract. Authors should solve this issue.

The sentence in the discussion was removed from the text and the sentence was rearranged.

# Comment 2: Line 169. Please authors, clarify this important sentence.

Added what the differences are.

Conclusion:

# Comment 1: Please, authors should consider to change conclusion section to be in line with their findings.

Conclusion section has been rearranged.

Best regards.

Round 2

Reviewer 3 Report

Dear Authors,

I would like to thanks your effort on answering all my queries and comments. After the review, the manuscripts have improve significantly. This manuscript could be considered for publication on its present form.